# Unveiling Latent Structure of Venture Capital Syndication Networks

**DOI:** 10.3390/e24101506

**Published:** 2022-10-21

**Authors:** Weiwei Gu, Ao Yang, Lingyun Lu, Ruiqi Li

**Affiliations:** UrbanNet Lab, College of Information Science and Technology, Beijing University of Chemical Technology, Beijing 100029, China

**Keywords:** syndication network, venture capital, network embedding, investment behaviors, objective classification

## Abstract

Venture capital (VC) is a form of private equity financing provided by VC institutions to startups with high growth potential due to innovative technology or novel business models but also high risks. To against uncertainties and benefit from mutual complementarity and sharing resources and information, making joint-investments with other VC institutions on the same startup are pervasive, which forms an ever-growing complex syndication network. Attaining objective classifications of VC institutions and revealing the latent structure of joint-investment behaviors between them can deepen our understanding of the VC industry and boost the healthy development of the market and economy. In this work, we devise an iterative Loubar method based on the Lorenz curve to make objective classification of VC institutions automatically, which does not require setting arbitrary thresholds and the number of categories. We further reveal distinct investment behaviors across categories, where the top-ranked group enters more industries and investment stages with a better performance. Through network embedding of joint investment relations, we unveil the existence of possible territories of top-ranked VC institutions, and the hidden structure of relations between VC institutions.

## 1. Introduction

Venture capital (VC) is typically defined as the investment made by professional institutions in unquoted startup companies with long-term financial growth potential, high risks but also possible enormous returns [1]. Over the past 30 years, VC has been an important financing source for early-stage and emerging companies with innovative technology or business models. Besides financial support, VC institutions can also provide a startup or young business with valuable sources of guidance, consultation, and bring tremendous benefits by tapping into domestic business communities via knowledge spillover effect [2]. China now receives over 40 percent of global VC investments [3], and more and more Chinese startups and urban economies are benefiting from VC investments [4,5]. A variety of VC-backed companies, such as Tencent, Alibaba, and Baidu, have had a huge influence on China and the worldwide economy.

Making joint-investments with other VC institutions, which forms an ever growing syndication network with complex structure, is a good way to share information [6], resources [7] and benefit from mutual complementarity [8], diversity [9,10] to against uncertainties [11,12,13,14].

VC institutions are bounded by their current and past syndication, i.e., joint investments with others. Besides identifying leading VC institutions, the syndication network of VC institutions can be used to predict the alliance’s exit [15], quantify the social capital, evaluate the reputation of VC institutions [16], and predict the portfolio failure rate [17]. Unveiling the latent structure of VC syndication networks and investment behaviors is crucial for deepening our understanding of the VC industry and boosting the healthy development of the market and economy. However, VC syndication network encodes nodal interactions but neglects other investment behavior information, such as the number of IPO (Initial Public Offering), M&A (Merge and Acquisition), and the number of investments, and hidden richer higher-order structure should be better revealed, all of which are important for VC analysis.

Previous studies indicate that there is a leader-follower phenomenon in the Chinese VC market, for example, a few leading VC institutions have better access to high-quality startups and they usually set up investment plans, while other VC institutions have a higher tendency to follow [18,19]. When young inexperienced VC institutions make joint investments with leading VC institutions of great reputation, they would have a smaller chance to renege or shirk [20], and usually can gain a better position for competition by building a word-of-mouth reputation connections [20,21]. And leading VC institutions also can act as bridges for resources and investment skills exchange, as they are usually hub nodes in the syndication network [19]. The ecosystem of VC market is way more complicated than a leader-follower structure, but most previous studies only focus on identifying leading VC institutions [22,23,24], and they usually do not have an objective criterion for identifying other groups of VC institutions. For example, Batjargal et al. [22] analyzed empirical data to reveal investment patterns associated with leading VC institutions. Huang et al. [23] found the triadic closure probability is higher if a leading VC institution is in it. Yet, the above works only used the terminology of “leading VC institutions”, but the classification is determined according to subjective criteria or practical experience from VC practitioners. Recently, a few works are dedicated to identifying leading VC institutions from joint-investment networks. The task is then equivalent to determine the most influential nodes via various centrality measurements in complex network analysis [25,26,27,28,29]. Yang et al. [24] employed a weighted k-means algorithm to identify leading VC institutions based on various centrality measurements. With the aid of graph convolutional deep learning model, Gu et al. [30] discovered that the attention matrix can be applied to rank VC institutions. However, most related works are based merely on network centrality measurements or partial nodal interactions, and these works pay more attention to leading VC institutions but do not make a classification of remaining institutions. Besides, they also neglect the richer higher-order structure and other investment behavior information such as IPO, M&A, and investment frequency, which hinders gaining a more comprehensive understanding of the VC market. To the best of our knowledge, few works are focusing on the objective classification of VC institutions and further ranking of obtained groups.

Starting a new VC institution is an activity that entails a high level of risk, both success, and failure cases are valuable. Failure is also an important characteristic of the entrepreneurial reality though it is also been viewed from a negative perspective [31]. In this paper, we propose an iterative Loubar method [32,33,34] based on Lorenz curve [35] to make objective classification of successful as well as unsuccessful VC institutions based on their principled features, which do not require setting arbitrary thresholds to separate different categories and the number of categories can be automatically determined. The features fed to our method include the number of IPO (Initial Public Offering)&MA (Merge and Acquisition) and the number of investments of VC institutions, which are important indicators of investment performance [12].

The systematical differences between positions in the VC syndication network and investment behaviors among VC institutions in each category prove the validity and practicability of our method. VC institutions in top-ranked categories generally enter more industries and get involved in more investment stages, have a higher network centrality, and have better investment performance. In addition, through the network embedding techniques [36], we unveil a latent structure of different categories of VC institutions, where top-ranked VC institutions have their territories. Our study can benefit on developing better investment strategies and portfolio optimization.

## 2. Data Preprocessing

In this paper, we conduct the study based on the SiMuTong database that records detailed information about each investment, including the VC institution that made the investment, the startup company that got invested, investment date, investment amount, currency, share, investment stage, and some basic information about the startup, for example, its industry and the location of its headquarter. We remove investment records without information about the investor or startup. And, for better consistency, we map the industry type of startup companies provided by Simutong to the ones in the Industrial Classification for National Economic Activities of China (ICNEAC). In addition, we convert the investment amount that was in foreign currency into RMB.

In the western world, the VC industry emerged in 1946 and experienced its first boom and bust cycle from 1982 to 1993 [12]. While in China, VC is still a relatively newly emergent industry (see Figure 1a). Before the 1990s, China received few VC investments. After 2000, The size of the annual investment amount and the number of investment activities underwent a rapid increase, and they both manifest a roughly five-year cycle (see Figure 1a). From 2014 to 2017, there was a spur of VC institutions that around four thousand new VC institutions entered the market (see Figure 1b), which further indicates the urgency of developing a quantitative science of VC institutions.

## 3. Iterative Loubar Algorithm for Objective Classification

In the VC literature, traditional ranking algorithms use either the Delphi method based on interviews with practitioners, which are usually limited on the sampling size and influenced by subjective experience, or the network-based centrality measures, which neglect higher-order structure. For example, for the Delphi method, the number of experts got interviewed is usually quite small, and it lacks an objective evaluation criterion and a clear conflict reconciling mechanism when two or more interviewed experts do not come to a consensus on the same question. Besides, both the Delphi and the centrality-based methods pay disproportionately more attention to leading VC institutions rather than underperforming ones, and they are unable to give an objective classification of VC institutions. In this paper, we devise an iterative Loubar method without setting arbitrary thresholds and the number of categories, both of which can be automatically determined.

The input evaluation indicators of VC institutions fed to the algorithm are the number of investments and the number of IPO&MA. VC institutions that can make a large number of investments usually have substantial funds, which is a manifestation of good financial condition and great social capital; while the number of investments exited through IPO and M&A is the most successful way of exit for VC institutions [12].

Our iterative Loubar method works as follows: First, ascendingly ordering all VC institutions according to their number of investment deals, accumulate their values on the vertical axis, and normalize both axes by the corresponding maximum (i.e., normalize the data by the cumulative number of investments for the vertical axis, and the number of VC institutions for the horizontal axis). This curve is also named the Lorenz curve [32] (see the lightest blue curve in Figure 2a). Then take the derivative of the Lorenz curve at (1,1) and extrapolate it to the point at which it intersects the horizontal axis, which gives us an objective threshold to separate the first level of VC institutions from others on this indicator. All VC institutions above this threshold will be classified as temporary level 1 on this indicator. The same process is repeated on the other indicator, here, the indicator is the number of IPO&MA (see Figure 2b), which will also classify some VC institutions into level 1 on this indicator. Eventually, only the VC institutions in both level 1 on the two indicators can be classified as the first category (named “first-tier”, see Figure 2c).

We then exclude VC institutions in first-tier from the Lorenz curve and re-normalize the remaining curve to repeat the previous process for both indicators to obtain VC institutions in level 2 (we name the second category as “second-tier”, see orange hexagon in Figure 2c), which would have the ones in both level 2 on the two indicators and the ones in level 1 on one indicator and in level 2 on the other indicator. If necessary, we can make three subdivisions for the second category. The algorithm goes on until no VC institution is left (see Figure 2a,b), and here we obtain five categories (see Figure 2c), which are also in meaningful ranking from first-tier to fifth-tier as we will illustrate later. By contrast, ordinary clustering methods cannot make ranking for obtained clusters. The Lorenz curves are depicted in progressively transparent shades of blue (from light to dark ones, corresponding to level 1 to level 5 in Figure 2a,b), and the corresponding derivatives at (1,1) from light red to dark red. Our method can be easily extended to situations with more input indicators. More details and discussions of the iterative Loubar algorithm can be found in Appendix A.

We eventually classified firms into five categories. Those leading VC institutions in the first category are labeled as “first-tier”, the ones in the following are categorized as “second-tier”, “third-tier”, “forth-tier”, and “fifth-tier”, respectively (see Figure 2c). The VC institutions classified into the first category (e.g., Starter Story, Statista, Sequoia Capital China, and China Growth Capital) by our method are in accordance with authority ranking reports, which rank VC institutions based on their capability in earning money, reputation, investment frequency and so on [37]. Compared to the first category, the ones in the second category (orange hexagon in Figure 2c) have fewer investments or IPO and M&A. This group also contains VC institutions with a large number of investments (in level 1 on this indicator) but fewer IPO and M&A (level 2 on the other indicator), and vice versa. The majority of VC institutions in the fifth category (“fifth-tier”, see purple hexagons in Figure 2c) have only a few investments and IPO and M&A exits, but there are some VC institutions that make a large number of investments but end up with quite a few successful exits, which generally have a worse investment performance and much less successful.

## 4. Investment Behaviors and Performance Analysis

We make further investigations on VC institutions of each category and discover that they have quite different investment behaviors. We first analyze the number of investment deals, IPO and M&A, investment stage, and investment industries for VC institutions in each category. During different investment stages, which consist of seed, initial, expansion, and maturity, startup companies need different resources or guidance from VC institutions. Thus the number of stages that a VC institution got involved in can reflect their ability on the whole investment cycle. While the number of industries that a VC enters is also a manifestation of their knowledge in various fields and the size of funding. As shown in Figure 3, VC institutions in the first category not only enter a wider range of industries but are also involved in more investment stages. There is a clear decrease in such indicators for VC institutions from the fi.

We then evaluate the success rate of VC institutions from different categories by dividing the number of IPO&MA by the total number of invested startups. As shown in Figure 4a, the IPO and M&A rate of VC institutions from the first category is even lower than the ones from second- to forth-tier. This is counter-intuitive since many VC ranking systems give higher ranks to VC institutions based on their high IPO rates [37,38]. Some evidence also suggests that leading VC institutions have a larger fraction of investments that exit through an IPO or M&A [39]. To better investigate the investment performance, we propose a new indicator named hawk eye index (HEI), which is defined as the amount of money invested into the startups that exit through IPO divided by the total amount of money of all investments. It is formulated as follows: $HEI=\left(Amt_{IPO}/Amt_{Total}\right)/\left(\#investments_{IPO}/\#investments_{Total}\right)$. As shown in Figure 4b, the median HEI of VC institutions in the first category ranks first and decreases over the remaining four tiers in order. A higher HEI indicates that the VC institution might try out for a larger number of investments, but have a better ability to allocate more resources on deals that are of greater potential and eventually successful.

In addition, VC institutions play an important role in the development of the economy, their survival rate and sustainable development are closely related to economic growth. Here, the survival rate is defined as the proportion of VC institutions that have yet to continue to invest after *t* years after their first investment. We calculate the survival rate of VC institutions from different categories obtained by our iterative Loubar classification method. We find that the longevity curves rank in decreasing order in accordance with ordered categories by our iterative Loubar classification algorithm that VC institutions from the first-tier have the highest survival rate and the fifth-tiers have the lowest (see Figure 5). Based on statistical analysis, Govindarajan et al. discovered that nearly 40% of companies cannot survive within the first 5 years [40], which is just an average for the whole market, but the whole system can be quite heterogeneous. From Figure 5, we can observe that over 95% of first-tier VC institutions can survive over 5 years, while more than 90% of fifth-tier VC institutions died in 5 years after their first investment. The variety of survival rates might be caused by the heterogeneity of VC institutions. The longevity survival curves, which rank in decreasing order for first-tier to fifth-tier VC institutions, also confirm the effectiveness of our iterative Loubar classification algorithm.

Syndication between VC institutions is a common investment strategy to share resources, access to better deal flows, reduce risks as well as complement skills [12]. In this paper, we analyze the difference in syndication behaviours of VC institutions from different categories. We first compute the number of joint investments $W_{ab}$ between VC institutions from categories *a* and *b*, and we then calculate the total number of joint-investments of each category $W_{a}=\sum_{b}W_{ab}$. Then the syndication tendency from category *a* to *b* can be represented as $W_{ab}/W_{a}$, which indicates that each row is subject to normalization, and it is worth noting that the matrix of $W_{ab}/W_{a}$ is asymmetric (see Figure 6).

From Figure 6, we can clearly observe that the VC institutions from the first-tier category syndicate more frequently with VC institutions from the same group than with VC institutions from other groups, which indicates a rich-club phenomenon. VC institutions in other categories except for the ones from the fifth-tier category also have the highest tendency to make joint investments with VC institutions from the first-tier category, which are indicated by darker color in the first column of Figure 6. VC institutions in the fifth-tier category, most of which are new entrants, have the strongest tendency to syndicate within its category. This might suggest that resources, better deal flows, and even information is more concentrated in those VC institutions from the first-tier category, and most VC institutions would like to mingle, but certain barriers exist. It is worth noting that VC institutions in every category have the lowest syndication tendency with VC institutions from the forth-tier category (see the fourth column in Figure 6), in comparison, the syndication tendency with VC institutions from the fifth-tier category is even much larger (see the fifth column in Figure 6), which is roughly at least twice higher than with the forth-tier category. These new VC institutions from the fifth-tier category might be more attractive to other VC institutions to collaborate with, but the forth-tier category might already pass such a new entrance phase and has less potential or resources to revive.

The syndication patterns can be better revealed through a complex network perspective. If two VC institutions make a joint investment, then we connect these two nodes in the network. We further compare the structural differences of VC institutions from different groups via computing the degree centrality, closeness centrality [41], and *k*-core centrality [42]. Degree centrality measures the number of edges that a VC institution has in the joint-investment network. With more ties, a VC institution usually has more investment opportunities and better deal flows [12]. Closeness measures its average distance to all other nodes that VC institutions with a high closeness centrality are averagely closer to other nodes [43]. The coreness (also referred to *k*-core) is a more sophisticated centrality measure, which can be obtained by iteratively removing vertices with a degree less than *k* [42] or by iterating the *H*-index operator [44]. The core-periphery structure indicates a higher-order organization in the VC syndication network. VC with a high *k*-core value is generally in a more central position with a stronger influence in the network [42]. We find that the VC institutions from different categories are of typically different values, and the first-tier category are more central than VC institutions from other categories (see Figure 7). From first-tier to fifth-tier VC institutions, there is a clear decreasing trend on all these network indicators, which also confirms the effectiveness of ranking for groups obtained by our iterative Loubar method. We further validate the quality of classification by our algorithm by comparing the structural differences with the algorithm proposed by Yang et al. [24], which uses a weighted k-means method to identify leading VC institutions and classify VC institutions into different groups. Recent evidence indicates that weighted k-means is a pretty good clustering algorithm for grouping VC institutions when compared with other clustering methods [24], and thus in this paper, we compare our classification results with the weighted k-means clustering algorithm. We discover that groups identified by the weighted k-means algorithm have no significant difference on all three network centrality indicators (see Figure 8). Since network centrality measurements are classical ways of quantifying how important a VC institution is, our algorithm has a better classification performance.

## 5. Latent Structure of Joint-Investment Behaviors

To further investigate the phenomena observed above, we exploit network embedding algorithms to better unveil the latent structure of VC institutions in the Chinese VC market. Our embedding method builds upon the DeepWalk architecture [45], which is a direct adaptation of the word2vec [46] model in the context of graphs. Random walk sequences on the syndication network, which are considered as “sentences” in the natural language processing, are fed to the SkipGram model [46] to obtain real-value vector representations for each VC institution. If two VC institutions are of a closer relationship in the syndication network, then they have a close relationship in the embedded metric space. The vector representation can be easily integrated into other computational models to make accurate predictions on joint investment behaviors and to cluster similar VC institutions. Since the embedding vectors of syndication also encode the complex network structure in the investment space, they can also help us have a better view of the evolution of the Chinese VC industry.

In this work, to get their vector representations, based on the accumulative joint-investment networks, we embed VC institutions into a 100-dimensional metric space, which is suggested by the recent advance on determining an appropriate embedding dimension [47]. To visualize the embedding vector, we use t-SNE [48] to project the high-dimensional vector representation into two dimensions. We visualize the syndication evolution process in Figure 9 and we can tell that there are only a few VC institutions that have joint investments before 2000. Compared with the VC institutions from other categories, first-tier and second-tier VC institutions entered the Chinese market at a relatively earlier time (see Figure 9a). But some first-movers might be less successful and fall into lower-ranked categories (see Figure 9a). In 2003, many first-tier VC institutions already take the central position (see Figure 9b), which indicates that their investments play a key role in the Chinese VC industry. The syndication around first-tier and second-tier VC institutions began forming densely connected communities. VC institutions with fewer syndication events are located far from the majority (See Figure 9b–d), among which we notice several first-tier VC institutions that entered the investment market for the first time. These VC institutions are sort of rising stars that started from the fringe of the network but ended up as a superstar with a central position. They gain good opportunities to syndicate with second-tier and first-tier VC institutions and gradually move to the densely connected elite community (see Figure 9c,d). It is worth noting that there are two separate groups in the Chinese market, one of which is mainly consisted of VC institutions from the forth-tier and fifth-tier categories, which are in accordance with the results presented in Figure 6. The fifth-tier group has a densely connected structure among them, so they might have limited opportunities to source high-quality deal flow (i.e., select promising startup companies) and to nurture investments (i.e., add value to portfolio companies) [12].

Network embedding not only encodes the syndication relationship between VC institutions but also provides a natural explanation for the architecture of real complex networks with a latent metric space. We measure the geometric properties of VC institutions by quantifying the number of nodes within a certain distance *R*.

As shown in Figure 10a, the average number of VC institutions that are in the vicinity of a first-tier VC institution varies significantly across categories. VC institutions in the first-tier category are more central as the average number in their vicinity grows faster when *R* is larger. We discover that there is a plateau for a first-tier VC institution to encounter other first-tier ones when *R* is around 1.2 to 1.5, that there is no clear significant increase. This suggests the existence of possible territories of first-tier VC institutions that generally a few of them would dominate a certain range, where no more other first-tier institutions can be encountered. We then set R=1.35, which resides in the plateau phase, and we find that first-tier VC institutions are surrounded by many fifth-tier VC institutions (indicated by a large fraction of entities in the fifth category, see the first value of the purple line in Figure 10b), while first-tier VC institutions themselves seem to avoid each other, which is indicated by a near-zero fraction. While fifth-tiers are either closer to first-tier VC institutions or closer to each other. When we look at a wider range, e.g., R=2 (see Figure 10c), first-tier VC institutions are still much closer to each other than the rest. This indicates that the first-tier VC institutions cooperate frequently, but they all have some smaller VC institutions that are only closer to themselves, which might form their own territories.

## 6. Conclusions

In this paper, we devise an iterative Loubar algorithm to make an objective classification of VC institutions and ranking obtained categories, which does not require setting arbitrary thresholds and the number of categories and can be easily extended to cases with more indicators. We discover that the average performances of VC institutions in identified categories, from the first-tier category to the fifth-iter, on various investment behaviors indicators, largely follow a descending order, which confirms the validity of our algorithm. VC institutions from top-ranked first-tier categories generally enter a wider range of investment industries, involve in more investment stages, as well as have a larger number of syndication with VC institutions from other categories. The only exception lies in the success rate, where those VC institutions in the first-tier category are not high, but we find that they have the highest hawk eye index, which indicates that they might make many tryouts but are able to allocate more resources into startups with great potential. This also suggests that the ordinary success rate, which equals the fractions of exits through IPO and M&A [12], is not a proper indicator to identify leading VC institutions. In addition, we find that most VC institutions have a stronger syndication tendency than VC institutions from the first-tier category. The exception is the fifth-tier category, which has a much stronger tendency to make joint-investment within its category. It is worth noting that VC institutions from every category have the lowest syndication tendency towards the VC institutions from the forth-tier category. The latent structure of the syndication network unveiled by network embedding further confirms our previous findings and suggests the existence of possible territories of first-tier VC institutions that generally a few of them would dominate a certain range, which is indicated by the plateau in Figure 10a, where no more other first-tier institutions can be encountered.

Our algorithm can be easily extended to make finer classifications. For example, in the second category, there are some VC institutions that have a larger number of investments that are identified as level 1 on this indicator and level 2 on the IPO&MA indicator (see those relatively scattered orange hexagons on the bottom-right in Figure 2c), such institutions might be less successful than those with the number of investments at level 2 but IPO&MA at level 1 (see those a few concentrated orange hexagons on the top-left in Figure 2c).

In the future, a closer investigation of the investment strategies of VC institutions in each category would be critical for developing a quantitative theory on the success of VC institutions. In this study, we focus our analysis on the Chinese market due to the limitation of data accessibility, which poses great challenges for making comparative analyses across countries.

## Figures and Tables

**Figure 1 entropy-24-01506-f001:**
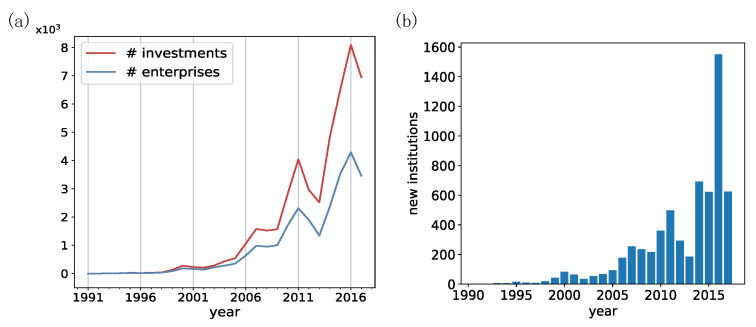
Basic statistics of VC investment activities in China: (**a**) The number of investments and the number of startups got invested in each year; (**b**) The number of new VC institutions that enter the Chinese market each year.

**Figure 2 entropy-24-01506-f002:**
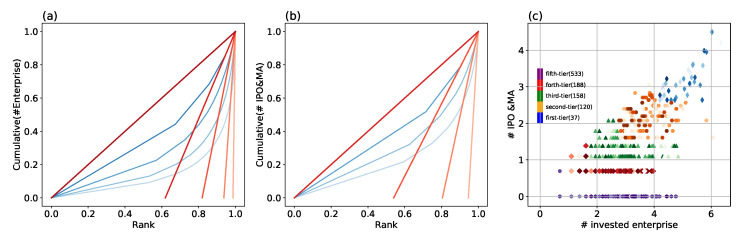
The illustration of our iterative Loubar method on classifying VC institutions: The partition based on (**a**) the number of investments, (**b**) the number of IPO and M&A. (**c**) the classification results are based on the number of investment deals and the number of IPO and M&A. The digit in the parentheses indicates the size of the category. The color of the symbols indicates the density of data points, a darker color corresponds to a higher density.

**Figure 3 entropy-24-01506-f003:**
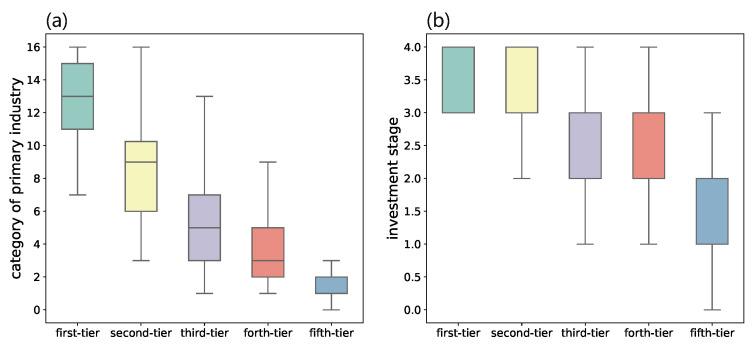
Comparisons between VC institutions from different categories on the number of (**a**) industry, and (**b**) investment stages that they involved.

**Figure 4 entropy-24-01506-f004:**
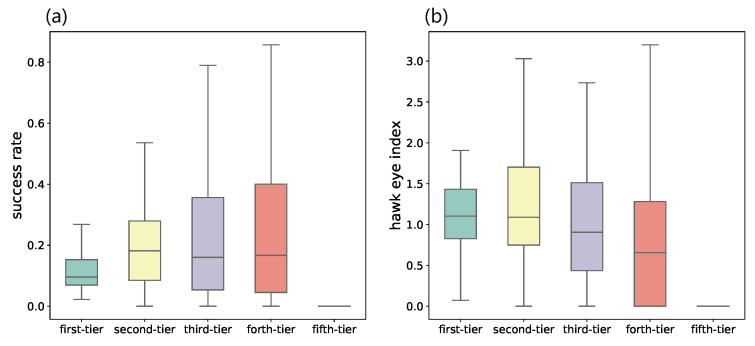
Comparisons between VC institutions from different categories on (**a**) the success rate, which is measured by the fraction of invested startups that exit through IPO or M&A, and (**b**) the hawk eye index (HEI). Note that the first-tier category has a low median success rate but the highest median HEI, which indicates that they tend to invest more money on eventually successful startups out of a large number of alternative choices.

**Figure 5 entropy-24-01506-f005:**
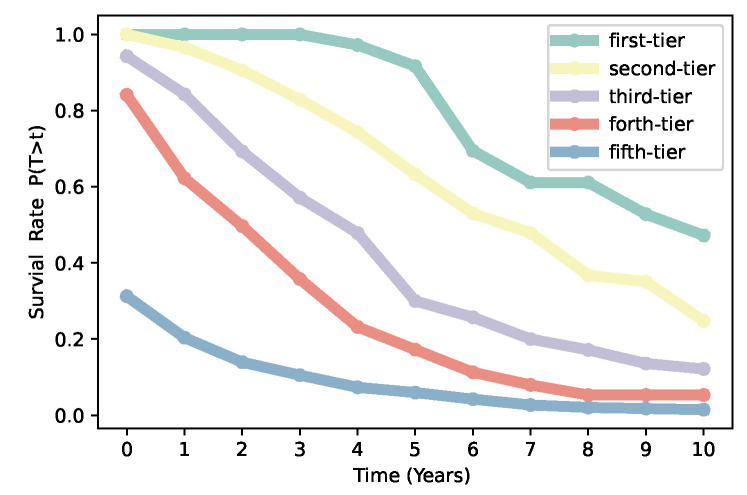
Comparisons between VC institutions from five categories on the survival rate.

**Figure 6 entropy-24-01506-f006:**
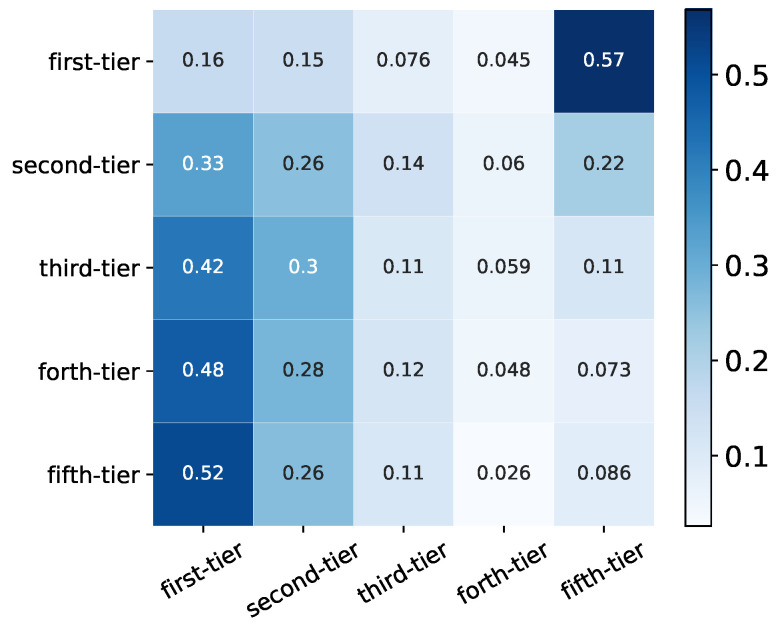
Syndication probability $W_{ab}/W_{a}$ of VC institutions from different categories. The first-tier group and the fifth-tier group have the highest tendency to make joint-investment with VC institutions within the same category. Each row is subject to normalization, but each column is not subject to normalization.

**Figure 7 entropy-24-01506-f007:**
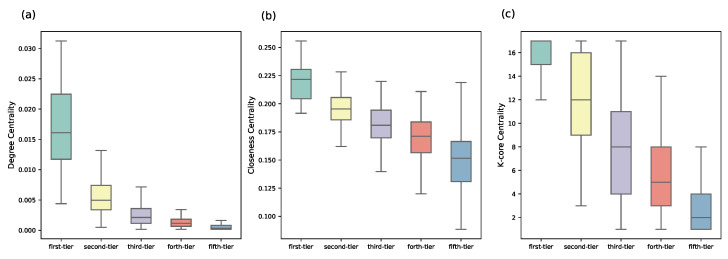
The difference between VC institutions of five categories identified by our iterative Loubar algorithm on three network centrality indicators: (**a**) degree; (**b**) closeness; and (**c**) k-core.

**Figure 8 entropy-24-01506-f008:**
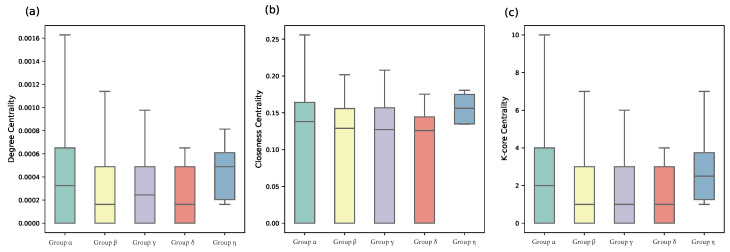
The difference between VC institutions of five categories identified by the weighted k-means algorithm [24], which cannot rank obtained clusters, on three network centrality indicators: (**a**) degree; (**b**) closeness; and (**c**) k-core. Note the α,β,γ,... only denote different groups but not the rank.

**Figure 9 entropy-24-01506-f009:**
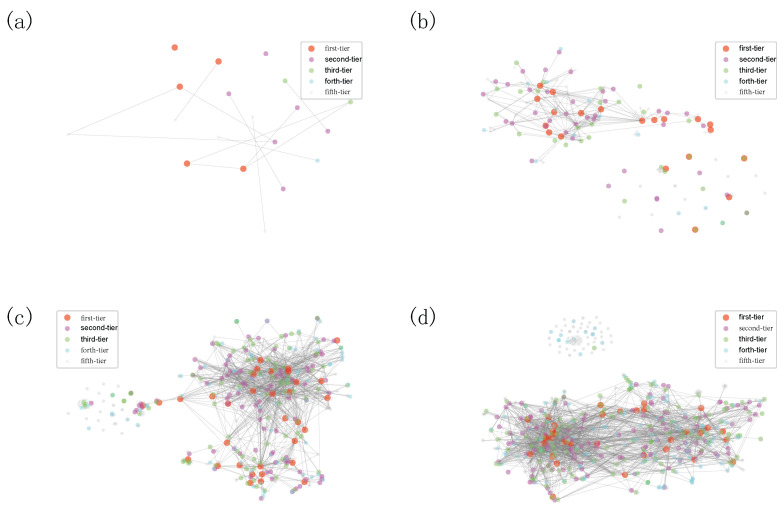
The two-dimensional projection of the cumulative syndication networks obtained by t-SNE algorithm in the year (**a**) 1998, (**b**) 2003, (**c**) 2008, and (**d**) 2011. Each node represents a VC institution with node color indicating the classification category obtained by our iterative Loubar method, and the gray line signifies the syndication relation between VC institutions.

**Figure 10 entropy-24-01506-f010:**
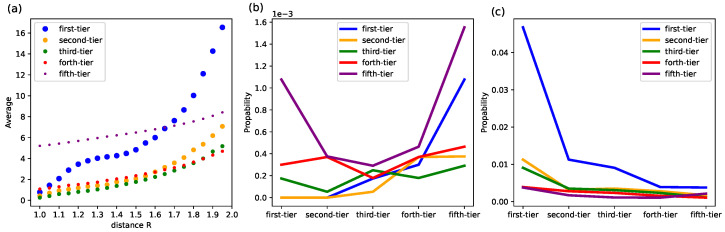
(**a**) The average number of VC institutions in the vicinity of a first-tier VC institution. The horizontal axis is the distance *R* in the embedded space and the vertical axis is the average number of VC institutions from different categories in the vicinity of a VC institution. The results are averaged for all VC institutions from the first-tier category. (**b**) The average fraction of VC institutions from different categories encountered in the vicinity with R=1.35 and (**c**) R=2 for VC institutions from different categories.

## Data Availability

The data on venture capital investments is purchased from SiMuTong dataset of Zero2IPO Group (www.pedata.cn) (accessed on 13 August 2018).

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
