# Peer review of "Unveiling Latent Structure of Venture Capital Syndication Networks"

_entropy, 2022, doi:10.3390/e24101506_

Round 1

Reviewer 1 Report

This paper deals with VC classification using Loubar methodology. This is not enough for a research paper. Why is this issue important? What are the questions that you are trying to answer? what is your hypothesis? You must link VC classification to other questions in order to make a significant contribution. For example, can the classification of VC predict its success rate?  Can better allocations of VCs improve economic development?   you can also compare the classification of VCs in other countries and say something about the role of government in VCs classifications and operation.

Author Response

Comments:

This paper deals with VC classification using Loubar methodology. This is not enough for a research paper. Why is this issue important?  What are the questions that you are trying to answer? what is your hypothesis?  

Response: Thanks for the comment, this motives us to better emphasize the contribution of our works. VC firms provide money, technical and managerial guidance or help to startup firms with long-term growth potential, and VC firms play an important role in the development of the economy. The whole VC market is of high diversity and quite heterogeneous and VC institutions can be quite different from each other, thus, classifying VC institutions are of great importance for gaining a better understanding of the investment behaviors of VC and better promoting the healthy development of the VC ecosystem. For example, only after detecting different groups, and then with help from network embedding techniques, we can reveal the territories of top-ranked VC institutions that have a closer distance with many other VC firms from other lower- ranked groups until they reach another big VC from the top-ranked group (see Figure 9 in the main text). 

More specifically, in this paper, our iterative Loubar method can objectively classify VC institutions into five categories without setting arbitrary thresholds. With the assumption that the universal investment laws and characteristics between the successful and less successful VCs are quite different, we analyzed the difference between VC institutions via analyzing their difference on a variety of investment behavioral indictors, including the investment industry and stage variety, investment syndication frequency, rate of IPO&MA, survival rate (which is newly added results), and social capital indicated by VC’s centrality measure (see Figures 3-7 in the main text).

we discovered that VC institutions from successful categories generally enter a wider range of investment industries, involve in more investment stages many tryouts but can put more resources into startups with great potential. This also suggests that the ordinary success rate, which equals the fractions of exits through IPO and M&A, is not a proper indicator to identify leading VC institutions. Through network embedding technology, we found the existence of possible territories of first-tier VC institutions that generally a few of them would dominate a certain range, where no more other excellent institutions can be encountered. Even the first-tier VC institutions cooperate frequently, but they all have some smaller VC institutions that are only closer to themselves, which form their territories, forming an elite-clique investment ecosystem. 

You must link VC classification to other questions in order to make a significant contribution. For example, can the classification of VC predict its success rate? 

Response: Thanks for the comment. The short answer is yes, the classification of VC can be used to predict the survival rate (newly added analysis shown in Figure 5), IPO&MA rate (Figure 4) and to predict its syndication preference (Figure 6), as well as social capitals which is indicated by nodal centrality.  

Ranking venture capitals and finding the elite venture capitals is an important hot topic in the field of VC studies, however, most of those algorithms are based on social capital analysis via node centrality measuring [1]. Our iterative Loubar method can do better, we can not only detect the top-ranked VC institutions but can give rankings to other detected groups, which exhibits descending trend on various indicators (see Figures 3-6 in the revised version).  

Can better allocations of VCs improve economic development? you can also compare the classification of VCs in other countries and say something about the role of government in VCs classifications and operation. 

Response: Thanks for the suggestion. We assume that better allocations of VC institutions can improve economic development, as many influential companies are backed up by VC institutions. But this question would require more economic data and a closer study in the future.  

       We also assume that the government may play an important role in the VC market, and when we start the project, we tried to make some comparative analyses across countries, but the limitation on data accessibility poses great challenges, VC data from other countries are very expensive. Once when we can obtain more data from other countries, such studies will be possible to study.  

References: 

[1] Yang, H., Luo, J. D., Fan, Y., & Zhu, L. (2020). Using weighted k-means to identify Chinese leading venture capital firms incorporating with centrality measures. Information Processing & Management, 57(2), 102083. 

Reviewer 2 Report

Thank you for giving me the opportunity to read your interesting paper!

I think, the methodological approach based on an iterative Loubar method/Lorenz curve to enable an objective classification of venture capital institutions is carefully chosen and purposeful. The results, i.e. that the latent structure of the syndicate network indicates the existence of possible areas of excellent and non-excellent VC institutions, are useful for research and  market evaluations and assessments.

My criticism relates more to the fact that the use of Lozenz curves is hardly ever critically considered and discussed, as it is gladly used in the economic and technical sciences for similar questions. However, it has been shown that the measurement of "concentration" (as a concept of descriptive statistics) raises many problems of understanding in relative terms:
1) which is understandable given the often superficial presentation of the Lorenz curve in the literature.

2) The problems of understanding relate primarily to the distinction between concentration and similar appearing terms such as dispersion, skewness, etc., which all have something to do with "inequality";

3) the distinction between absolute and relative concentration as well as the
interpretation of the content of "concentration characteristic" and "units" among which the sum of the characteristics is distributed; and

4) the interpretation of the limiting states of minimum and maximum concentration.

As interesting as the subject matter is and as informative as the results seem to be, there is no serious discussion of the potentials and limitations of the methodology used.

Author Response

Comments:

 Thank you for giving me the opportunity to read your interesting paper! 

 I think, the methodological approach based on an iterative Loubar method/Lorenz curve to enable an objective classification of venture capital institutions is carefully chosen and purposeful. The results, i.e. that the latent structure of the syndicate network indicates the existence of possible areas of excellent and non-excellent VC institutions, are useful for research and market evaluations and assessments. 

 My criticism relates more to the fact that the use of Lozenz curves is hardly ever critically considered and discussed, as it is gladly used in the economic and technical sciences for similar questions. However, it has been shown that the measurement of "concentration" (as a concept of descriptive statistics) raises many problems of understanding in relative terms: 

1) which is understandable given the often superficial presentation of the Lorenz curve in the literature. 

Response: Thanks a lot for the positive comments on our manuscript. We agree that the Lorenz curves should be used with deeper considerations. In this paper, we studied the advantages and limitations of the iterative Loubar algorithm over distributions with different heterogeneity. We find that the iterative Loubar algorithm generally works well on heterogeneous distribution (e.g., scale-free distribution with  ), but not on normal distribution or some special cases (e.g., uniform distribution). We added related analysis in the appendix.   

2) The problems of understanding relate primarily to the distinction between concentration and similar appearing terms such as dispersion, skewness, etc., which all have something to do with "inequality"; 

 Response: Thanks for the great suggestion. To our understanding, the Lorenz curve can reflect inequality, which relates to other concepts, including dispersion and skewness. Usually, a skewed distribution (e.g., a power law or named scale-free distribution ) will lead to a more curly Lorenz curve that reflects a stronger inequality. When the data is uniform (e.g., N entities have values from 1 to N, respectively), from the cluster analysis, they can be clustered into N groups, or just one group (as they can be treated as a manifold), but the iterative Loubar method cannot obtain “correct” classification on this special case, which is also a limitation of the method.

3) the distinction between absolute and relative concentration as well as the interpretation of the content of "concentration characteristic" and "units" among which the sum of the characteristics is distributed; and the interpretation of the limiting states of minimum and maximum concentration. 

As interesting as the subject matter is and as informative as the results seem to be, there is no serious discussion of the potential and limitations of the methodology used. 

 Response: Thanks for the great suggestions. When the data is homogeneous (i.e., every entity is of the same value), the Loubar method can correctly classify the whole population into just one category. When the population is quite heterogeneous (e.g., their values are from 1 to N), then the partition results are more consistent with common sense. When the data is strictly homogeneous (i.e., every entity is of the same value), then the method can correctly identify just one group. However, when the distribution is uniform or closer to a narrow normal distribution, it doesn’t work well as we pointed out before. The merits of the method lie in its objective nature that does not require setting any arbitrary thresholds, however, the partition results may still be affected by the heterogeneity of the data per se, which still requires a closer and more detailed study in the future. Please see more detailed discussions in our newly added Appendix in the main text.

Reviewer 3 Report

Very interesting and a novel paper. The use of Loubar algorithm to classify VP organizations is quite new. I suggest the authors include an appendix about the Lorenz curve and Loubar method. Some basic references are:
1. Louail, T., Lenormand, M., Cantu Ros, O. et al. From mobile phone data to the spatial structure of cities. Sci Rep 4, 5276 (2014). https://doi.org/10.1038/srep05276 (Ref. 30)
2. M. Beach and S. F. Kaliski , Lorenz Curve Inference with Sample Weights: An Application to the Distribution of Unemployment Experience, Journal of the Royal Statistical Society. Series C (Applied Statistics)Vol. 35, No. 1 (1986), pp. 38-45 (8 pages)
3. Rémi Louf Wandering in cities: a statistical physics approach to urban theory, November 2015, https://www.researchgate.net/publication/285271028_Wandering_in_cities_a_statistical_physics_approach_to_urban_theory#fullTextFileContent

Author Response

Very interesting and a novel paper. The use of Loubar algorithm to classify VP organizations is quite new. I suggest the authors include an appendix about the Lorenz curve and Loubar method. Some basic references are: 

  1. Louail, T., Lenormand, M., Cantu Ros, O. et al. From mobile phone data to the spatial structure of cities. Sci Rep 4, 5276 (2014). https://doi.org/10.1038/srep05276 (Ref. 30)
  2. M. Beach and S. F. Kaliski , Lorenz Curve Inference with Sample Weights: An Application to the Distribution of Unemployment Experience, Journal of the Royal Statistical Society. Series C (Applied Statistics)Vol. 35, No. 1 (1986), pp. 38-45 (8 pages)
  3. Rémi Louf Wandering in cities: a statistical physics approach to urban theory, November 2015,

Response: Thanks for your great suggestions, we have added more details of the Lorenz curve and Loubar method in the Appendix (from Line 372 to 448) with the recommended references.  

Reviewer 4 Report

The paper has three drawbacks:

- it is just illustration  of well-known statistical methods, no new methiodologiucal contribution;

- it limits the reserach to Chinese market (also old data), without any comparison to the other markets, which means not so much interest to the international readers;

- it does not show the reference and comparison to plenty of clustering methods proposed in the literature;

- it does not refer to scientific results related to VC market known in the area of economics and finance.

Author Response

Comments: 

The paper has three drawbacks: 

- it is just illustration of well-known statistical methods, no new methodological contribution; 

Response: We appreciate the reviewer’s insightful comment that prompts us to clarify the main contributions of our paper. Our main methodological contribution lies in classifying VC institutions by taking into account more than one factor and grouping them in an iterative way that yields meaning ranking of obtained categories. The previous Loubar method either makes binary partition [1] or apply only one factor to classify objects [2]. Iteratively applying the Loubar method to the main evaluation indicators gives us more objective classification results without the need to resort to arbitrary thresholds. Besides, our method gives meaningful ranking of obtained clusters. By contrast, the traditional Loubar method [1,2] deals with only one factor, however, there can be more than one principal factor in most grouping tasks, in our case, the number of investments (that quantifies financial condition, great social capital and better accessibility of startups) and the IPO and M&A numbers (that capture the investment quality) both matters for measuring VC institution’s success. In addition, our iterative Loubar method is very flexible and can be extended to grouping tasks with more principal indicators without setting arbitrary thresholds and the number of categories.  

References: 

[1] Louail, T., Lenormand, M., Cantu Ros, O. G., et al. (2014). From mobile phone data to the spatial structure of cities. Scientific Reports, 4(1), 1-12.  

[2] Bassolas, A., Barbosa-Filho, H., Dickinson, B., et al. (2019). Hierarchical organization of urban mobility and its connection with city livability. Nature Communications, 10(1), 1-10.  

- it limits the research to Chinese market (also old data), without any comparison to the other markets, which means not so much interest to the international readers;  

Response: Thanks for the suggestion. We like to conduct comparative analysis across countries, however, data accessibility is the biggest obstacle on the way. Due to access limitation to venture capital investment data, most classical and important articles in the field of venture capital is limited to a domestic market of a certain country, including the classical paper “whom you know matters” [3], which found that better-networked VC firms experience significantly better fund performance, and the famous book “The Venture Capital Revolution” wrote by Josh Lerner [4] are all only study U.S.-based VC institutions and didn’t have comparisons to other markets. Still, these works are valuable to the literature and of interest to international readers outside the USA.  

Similarly, results for the Chinese VC market can be useful to international readers who have interest in the Chinese VC market or want to make comparative studies. As for us, we are now cooperating with a VC firm and may be able to obtain the CrunchBase database, if so, we are also planning to conduct a comparative analysis across countries in the future.  

In the discussion part we added, “In this study, we focus our analysis on the Chinese market due to the limitation of data accessibility, which poses great challenges for making comparative analyses across countries.” 

References: 

[3] Hochberg, Y. V., Ljungqvist, A., & Lu, Y. (2007). Whom you know matters: Venture capital networks and investment performance. The Journal of Finance, 62(1), 251-301. 

[4] Gompers, P., & Lerner, J. (2001). The venture capital revolution. Journal of Economic Perspectives, 15(2), 145-168. 

- it does not show the reference and comparison to plenty of clustering methods proposed in the literature; 

Response: Thanks for the comment. In our paper, we compared with other methods, though not plenty, the reasons are twofold:  

1) our iterative Loubar classification method is different from ordinary clustering algorithms: clustering algorithms cannot rank obtained clusters, but our method can. In this paper, our iterative Loubar method classifies VC institutions into five ranked tiers from top to bottom: ‘first-tier, ‘second-tier, ‘third-tier, ‘forth-tier, and ‘fifth-tier. Based on the previous finding in the literature that better-networked VC firms experience significantly better fund performance [3], in this work, we use degree centrality, closeness centrality, and k-core centrality to quantify VC’s network position and evaluate VC’s performance. As shown in Figure 6, the VC institutions in the first-tier category (Top-ranked group) are more central than institutions from other groups. In comparison, the weighted k-means algorithms, as shown in Figure 7 in the main text, it cannot distinguish performance differences between different clusters.  

2) In page 3 of Ref[5], the authors stated that “In contrast, the k-means algorithm cannot rank observations and also depends on the number of clusters”, the clustering method is not capable of performing the ranking task. Besides, In Ref[5], the authors have already made a comprehensive comparison between the weighted k-means, with which we compare, and many other popular clustering algorithms, such as TOPSIS (see Table 3 i Ref[5]) and the weighted k-means achieved the best clustering performance among all other clustering algorithms.  

Thus, here, we only report the comparisons with the weighted k-means clustering method. In the revised paper, we added the following statement: “Recent evidence indicates that weighted k-means is a pretty good clustering algorithm for grouping VC institutions when compared with other clustering methods [5] (Ref[24] in the revised version), and in this paper, we compare our results with the weighted k-means clustering algorithm.” 

References: 

[3] Hochberg, Y. V., Ljungqvist, A., & Lu, Y. (2007). Whom you know matters: Venture capital networks and investment performance. The Journal of Finance, 62(1), 251-301. 

[5] Yang, H., Luo, J. D., Fan, Y., & Zhu, L. (2020). Using weighted k-means to identify Chinese leading venture capital firms incorporating with centrality measures. Information Processing & Management, 57(2), 102083. 

- it does not refer to scientific results related to VC market known in the area of economics and finance 

Response: Thanks for the great suggestion. We have emphasized the connections between our results and references to VC studies in the area of economics and finance in our revised manuscript. 

From figure 5, we found that the VC institutions from the first-tier category (i.e., top-ranked group) syndicate more frequently with VC institutions from the same group than with other groups. This discovery is in accordance with a paper published in Social Networks [6] that applied multi-agent modeling to prove that elite VCs prefer syndicating with other elite VC institutions to share high-quality resources and reduce risks. 

Better-networked VC firms experience significantly better fund performance and have higher prestige [3], the iterative Loubar algorithm can identify VC institutions with central network positions (Figure 6 in the main text) and rank venture capitals’ prestige accordingly. 

In this paper, we discovered that VC institutions from the top-ranked group (named as “first-tier”) by our iterative Loubar method may have a lower rate of IPO and M&A than the second-ranked or third-ranked, and even fourth-ranked groups (see Figure 4a in the main text), This is counter-intuitive, and suggests that IPO and M&A rate might not be appropriate to measure VC institutions’ prestige, which is commonly adopted in the literature or VC industry [7-9].  

Inspired by the reviewer’s insightful comment, we also added the following analysis to our revised manuscript to relate our study to the finding in the area of economics and finance, in Ref[10], the authors stated that nearly 40% of companies cannot survive within the first 5 years. However, we find this conclusion may vary based on the heterogenous of VC institutions and the survival rate is highly correlated with our classification results. From Figure 1, we can tell over 95% of first-tier VC institutions can survive over 5 years while more than 90% of fifth-tier VC institutions died in their first 5 investment years. The longevity survival curves that rank in decreasing order in also accordance with our iterative Loubar classification result from first-tier to fifth-tier. In Figure 1, we define the survival rate as the proportion of VC institutions that have yet to continue to invest after t years after their first investment.  

See Figure 5 in the revised version or in the attached PDF file. Caption of Figure 5: Comparisons between the survival rate of VC institutions of five groups.  

References: 

[3] Hochberg, Y. V., Ljungqvist, A., & Lu, Y. (2007). Whom you know matters: Venture capital networks and investment performance. The Journal of Finance, 62(1), 251-301. 

[6] Gu, W., Luo, J.D. & Liu, J. (2019). Exploring small-world network with an elite-clique: Bringing embeddedness theory into the dynamic evolution of a venture capital network. Social Networks, 57, 70-81. 

[7] Chou, T. K., Cheng, J. C., & Chien, C. C. (2013). How useful is venture capital prestige? Evidence from IPO survivability. Small Business Economics, 40(4), 843-863. 

[8] These are the Top Venture Capital Companies in China (2021). https://welpmagazine.com/these-are-the-top-venture-capital- 418 

companies-in-china-2021. Accessed: 2022-08-23. 

[9] Jain, A. These are the Top Venture Capital Firms of 2020. https://www.entrepreneur.com/article/361514. Accessed: 2022-08-23 

[10] Govindarajan, V., & Srivastava, A. (2016). Strategy when creative destruction accelerates. Tuck School of Business Working Paper, (2836135). 

Round 2

Reviewer 1 Report

The paper has been improved and also the motivation is now clearer

It can be accepted in the present form

Reviewer 2 Report

The authors have adequately taken up the critical reviews as well as the suggestions and made a better classification and discussion of the methodology used.

Reviewer 4 Report

The revised version contains only very minor changes. The main problem is the lack of possible generalization opportunities (one method, one market) which is important for papers published in widely respected journals. It might be appropriate to send the paper to the journals directed into Chinese VC market.